# The Corrosion of Stainless Steel Made by Additive Manufacturing: A Review

Gyeongbin Ko [1,†], Wooseok Kim [1,†], Kyungjung Kwon [1,*] and Tae-Kyu Lee [2,*]

1   Department of Energy & Mineral Resources Engineering, Sejong University, Seoul 05006, Korea; radsfl10@gmail.com (G.K.); kevin5588@naver.com (W.K.)
2   Department of Mechanical & Materials Engineering, Portland State University, Portland, OR 97201, USA
*   Correspondence: kjkwon@sejong.ac.kr (K.K.); taeklee@pdx.edu (T.-K.L.); Tel.: +82-2-3408-3947 (K.K.); +1-503-725-4293 (T.-K.L.)
†   These authors contributed equally to this work.

**Abstract:** The advantages of additive manufacturing (AM) of metals over traditional manufacturing methods have triggered many relevant studies comparing the mechanical properties, corrosion behavior, and microstructure of metals produced by AM or traditional manufacturing methods. This review focuses exclusively on the corrosion property of AM-fabricated stainless steel by comprehensively analyzing the relevant literature. The principles of various AM processes, which have been adopted in the corrosion study of stainless steel, and the corrosion behaviors of stainless steel depending on the AM process, the stainless steel type, and the corrosion environment are summarized. In this comprehensive analysis of relevant literature, we extract dominant experimental factors and the most relevant properties affecting the corrosion of AM-fabricated stainless steel. In selective laser melting, the effects of the scan speed, laser power, energy density, and the post-treatment technologies are usually investigated. In direct laser deposition, the most relevant papers focused on the effect of heat treatments on passive films and the Cr content. There has been no specific trend in the corrosion study of stainless steel that is fabricated by other AM processes, such as wire arc additive manufacturing. Given the rising utilization of AM-produced metal parts, the corrosion issue will be more important in the future, and this review should provide a worthwhile basis for future works.

**Keywords:** additive manufacturing; stainless steel; corrosion; selective laser melting; direct laser deposition; wire arc additive manufacturing

## 1. Introduction

Additive manufacturing (AM), which is commonly known as 3D printing, is a rapidly growing technique in a variety of fields where digital 3D design data is used to directly build up a product in a layer-by-layer process. AM refers to a variety of processes in which material is deposited, joined, or solidified using power sources such as a laser, an electron beam, or plasma [1]. The advantages of AM over traditional manufacturing methods are that very complex parts can be fabricated directly from a CAD model with minimal waste generation, using less time. These advantages are sufficient for many researchers to study AM to increase the efficiency of sophisticated operations, and its market value is expected to reach $21 billion by the end of 2020 [2]. Therefore, many studies are ongoing, which compare the mechanical properties, corrosion behavior, and microstructure of the metals that are produced by AM or traditional manufacturing methods.

Stainless steels (SS) are iron-based alloys that typically contain Cr that is resistant to corrosion, and they are categorized by the differences in composition. A representative stainless steel is 316L, with the following alloy chemistry: 16–18.5% Cr, 10–14% Ni, 2–3% Mo, <2% Mn, <1% Si, <0.045% P, <0.03% C, <0.03% S, and balance Fe. A lower carbon content in 316L is particularly advantageous for welding processes to prevent carbide precipitation at grain boundaries. As a result, 316L is widely employed in marine engineering, pharmaceutical

manufacturing, potable water systems, and medical implants. The last application particularly has grown in recent years, and 316L can now be used in cardiovascular implants, orthopedic bone fixation devices, orthodontic materials in craniofacial applications, and artificial eardrums [1]. Martensitic precipitation hardened (PH) SS, which includes 17-4 PH and 15-5 PH, usually possess a ductile martensitic matrix that achieves significantly high strength through nanometric precipitates. Due to its enhanced mechanical properties and excellent corrosion resistance, martensitic PH SS are commonly used in the aerospace industry, petrochemical plants, and nuclear reactors among many other applications [3]. In addition, various SS, such as 304L and AISI 420 SS, are used as materials for AM.

This review focuses exclusively on the corrosion property of AM-fabricated SS by comprehensively analyzing the relevant literature, which is unlike other review papers that covered the process parameters, microstructure, and mechanical properties of steel [4]. In this paper, the principles of various AM processes, which have been adopted in the corrosion study of SS, and the corrosion behaviors of SS that depend on the AM process, the SS type, and the corrosion environment are summarized. In this comprehensive analysis of the relevant literature, we extracted the dominant experimental factors and the most relevant properties that affect the corrosion of AM-fabricated SS. Given the rising utilization of AM of metal parts, corrosion protection and durability will play an increasingly important role in the future, and this review could provide a worthwhile basis for future works.

## 2. AM Process Review

### 2.1. Powder Bed Fusion

Powder bed fusion (PBF) is the most advanced type of 3D printing, and this technology uses a high-power energy source, such as a laser or an electron beam, to selectively melt powder. The PBF process involves selectively fusing or sintering regions of a powder bed, which can either be plastic or metal in nature. The PBF technology often requires time and iterations of significant post-processing to yield a successful part, and significant training is necessary to obtain a successful part. Moreover, swapping materials requires a lot of labor hours. Nevertheless, PBF technology is in the limelight, because it can create precise and intricate geometries. It is also possible to automate production. Examples of the powders that are used in PBF include metals, ceramics, polymers, composites, and hybrids, which can be typically recycled [5]. The PBF process requires a roller or a blade to continuously spread material over the previously printed layers, and a sealed vacuum environment is required in the case of using an electron beam. The technology includes multi-jet fusion, selective laser sintering (SLS), selective laser melting/direct metal laser sintering (SLM/DMLS), and electron beam melting (EBM). SLS and SLM are the main examples of powder-based 3D printing processes. The SLS process was developed by Carl Deckard and Joseph Beaman in 1987, and it is an upgraded version of a liquid curing 3D printing technique. The SLS process can produce parts at a fast speed, and has high accuracy and variety of surface finishes. In this process, the surface quality and the strength of fabricated parts are mainly affected by the laser power, the temperature, and the part orientation. When SLS is utilized to manufacture metal products, the process is customarily referred to as SLM or DMLS, and the only difference between SLS and SLM is that the samples manufactured by SLM are sufficiently heated to the melting point [6]. The underlying solidified material layer and the additional powder layer are completely and partially melted and connected due to high-power energy.

### 2.2. Direct Energy Deposition

As a more complex AM process, direct energy deposition (DED) uses focused thermal energy in the form of a laser, a plasma arc, or an electron beam to continuously fuse materials by melting during deposition. It is commonly used to repair or add additional materials to existing parts, which is similar to welding, with extremely large build volumes. A typical apparatus consists of a multi-arm mounted nozzle, which deposits materials onto

specific regions of a surface for solidification. The nozzle is not axially restricted, so it has a more freeform pathway of production, but DED is only currently used for metals that are either wire or powder fed. DED technology includes direct laser deposition (DLD), laser engineering net shape (LENS), wire arc additive manufacturing (WAAM), laser metal deposition (LMD), and droplet-based 3D printing.

During the DLD process, a laser beam is applied to create a melt pool, and metal powder is carried and blown into the melt pool by inert gases. After the laser moves away, the melted powder joins and cools down to form a solid layer. The laser travels according to a toolpath for each layer. By repeating this procedure for each layer, a 3D part can be constructed [7]. DLD is an attractive technology for the rapid manufacture, modification, and repair of metallic components. It has been widely used for a long time for the deposition of wear- and corrosion-resistant coatings on mechanical parts and tools with materials that have excellent comprehensive properties [8].

WAAM is a DED method that combines arc welding and wire feedstock materials for an AM purpose. WAAM shows some disadvantages when compared to SLM or LMD. WAAM heat input is high, which induces high residual stresses and distortion, and the manufactured part's accuracy is therefore lower than in other AM technologies; WAAM always needs a finishing machining step as a result. Moreover, the low accuracy makes part redesigning difficult by using geometry optimization algorithms. On the other hand, WAAM has advantages, such as lower costs of welding equipment, wide availability of standard wires, virtually unlimited part sizes, and higher deposition rates. Therefore, the WAAM process is adequate for manufacturing preforms of large-size parts, which are manufactured from high-cost materials and require high volumes of machining [9]. Recently, droplet-based 3D printing that uses uniform metal micro-droplets as basic building blocks has been employed [10–13].

## 3. Corrosion Behaviors Depending on the AM Process, the SS Type, and the Corrosion Environment

### 3.1. Corrosion Behaviors of SLM SS

3.1.1. Corrosion Behaviors of SLM 316L SS

Corrosion Behaviors of SLM 316 SS in NaCl Solutions

The SLM method is one of the most widely commercialized technologies. Therefore, there were many papers regarding the corrosion properties of SLM-prepared SS. Corrosion research thus far has been conducted in terms of heat treatment, direction, and porosity in an NaCl environment. Sun et al. studied the effects of the scan speed and porosity on the corrosion of 316L SS to investigate a tendency that corrosion rate increases as porosity increases [14]. Figure 1 shows that an increase in porosity did not make a difference on the corrosion of SLM 316L. Meanwhile, they discovered that the corrosion behavior of SLM 316L was almost similar to that of wrought 316L, and the breakdown and repassivaton potential of SLM 316L decreased as porosity increased. The corrosion behavior of SLM 316L was investigated depending on the various scan speeds and the laser power in potentiodynamic polarization (PDP) tests by Sander et al. [15]. The corrosion potential and the current of SLM 316L were similar to wrought 316L, and they suggested that the scan speed and the laser power did not play a significant role in corrosion, but the resulting increase in porosity reduced the repassivation potential, which is shown in Figure 2. Murkute et al. established a correlation between the scan speed, ranging from 100 to 400 mm/s, and the corrosion properties, which determined that the higher the scan speed, the lower the corrosion potential and resulting corrosion resistance became [16]. When compared with the wrought samples, the scanned sample at a rate of 100 mm/s exhibited similar corrosion properties. In addition, Lin et al. demonstrated correlations between scan speeds, which ranged from 400 to 800 mm/s, corrosion properties, and density [17]. A sample at 500 mm/s showed maximum corrosion resistance, and a sample at 800 mm/s had the lowest density and corrosion resistance, with the roughest surface. The corrosion potential of the samples and the content of the grains with the orientations of (200) and

(111), which depended on the distance from the bottom, were measured using PDP and X-ray diffraction (XRD), respectively. The corrosion potential and the content of the grains with the orientation of (111) were higher, but the content of grains (200) and the grain diameter were smaller as the distance was longer. The authors revealed that more closely-packed crystal planes (111) and reduced grain sizes of the SLM samples led to improved corrosion behavior.

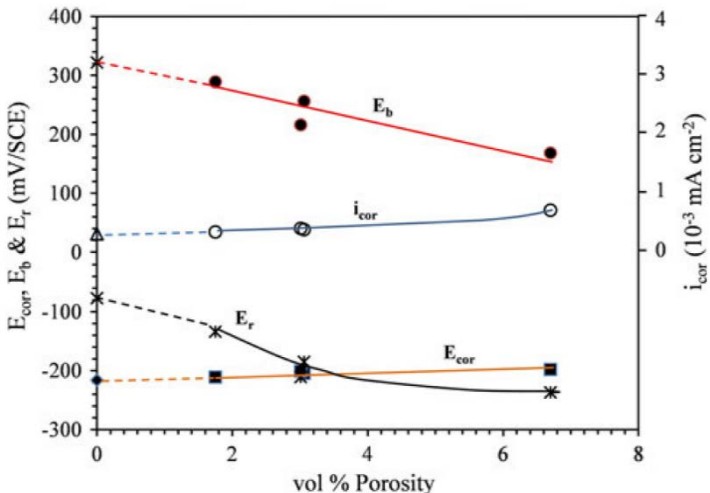

**Figure 1.** Corrosion potential ($E_{cor}$), breakdown potential ($E_b$), repassivation potential ($E_r$), and corrosion current density ($i_{cor}$) as a function of vol.% porosity in SLM samples [14].

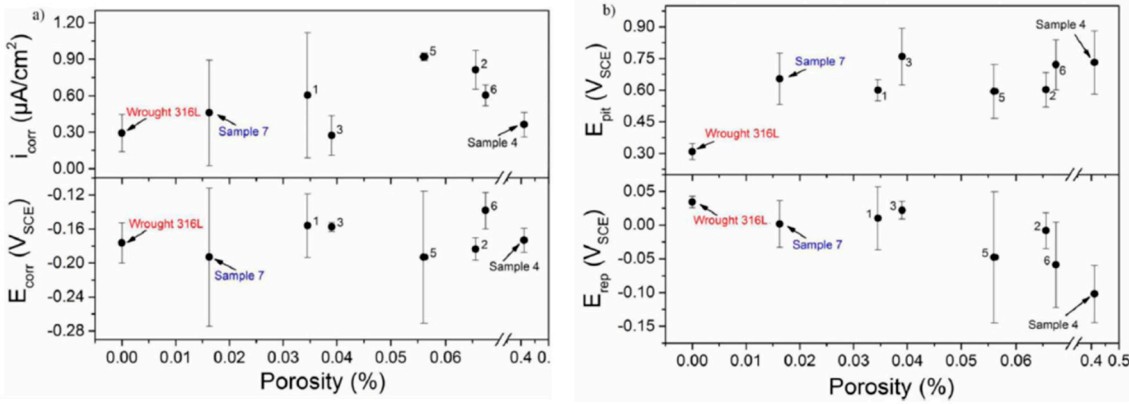

**Figure 2.** (**a**) $E_{corr}$, and $i_{corr}$, and (**b**) $E_{pit}$ and $E_{rep}$ values of the SLM 316L samples plotted against the porosity of the samples [15].

Zaharia et al. examined the corrosion behavior of 316L honeycomb cores by measuring actual weight loss without the electrochemical corrosion tests [18]. They adopted an accelerated experiment where 5% and 10% concentrated NaCl solutions were used, which allowed them to infer the corrosion behavior for five and a half years from a 10-day test. Chao et al. studied the effects of the MnS content and the presence of the Cr depletion zones on the pitting corrosion by transmission electron microscope coupled with energy-dispersive X-ray spectroscopy (EDS) element mapping and PDP tests, and they compared the SLM 316L samples to the wrought samples [19]. It was revealed that there were definitive 200–500 nm wide Cr depletion zones that surrounded the MnS inclusion, which reduced the pitting resistance. They discovered that the Cr depletion did not occur where the S element was absent by element mapping. The PDP tests showed that the pitting potential of the SLM 316L samples was higher than that of the wrought samples. The corrosion behaviors of SLM 316L and the wrought samples were investigated before and

after heat treatment by Ettefagh et al. [20]. The corrosion potentials of both samples increased after the heat treatments, and this difference of corrosion potential was attributed to the elimination of the residual stress after the heat treatment. Yusuf et al. first studied the effect of high-pressure torsion (HPT) plastic deformation on the microstructure and the corrosion behavior of SLM 316L, and they showed that the cellular structure and the voids disappeared when plastic deformation was given [21]. It was confirmed that the corrosion potential increased as the number of HPT plastic deformation increased, which would imply that HPT processing can generally improve the corrosion resistance of SLM 316L. Sun et al. studied the corrosion behavior of the crystallographic lamellar microstructure, which is where two differentially oriented grains (the major (011) grains and the minor (001) grains) appeared alternately in the 316L samples under lower laser energy density [22]. The lamellar microstructure displayed a very high breakdown potential compared to the commercially obtained 316L samples by the PDP tests. The alternately oriented grains based on the XRD analysis was regarded as one of the reasons for this behavior. The corrosion behavior of SLM 316L with additional TiC, which involved TiC/316L whose wt.% was 0, 2, and 4, was investigated by Zhao et al. [23]. The lowest passivation current among the TiC/316L samples was shown by 2 wt.% TiC/316L, and the passivation current of 4 wt.% TiC/316L was higher than that of 0 wt.% TiC/316L. The authors concluded that the addition of TiC would enhance the passivation of TiC/316L, but excessive TiC increased the corrosion rate of TiC/316L. Laleh et al. studied the erosion–corrosion properties of commercial and SLM samples by PDP and jet impingement measurement [24]. They showed the relatively higher breakdown potential and the lower repassivation potential of the SLM samples, as opposed to the commercial samples by PDP. On the other hand, the jet impingement measurement of SLM 316L showed lower corrosion resistance than the commercial sample. The presence of the induced pores within the SLM samples was assumed to lead to the low repassivation ability in the jet impingement test despite the low repassivation potential, and the low repassivation ability could be attributed to the low erosion–corrosion resistance on the SLM 316L samples. Duan et al. studied the effect of the solution aggressiveness on the corrosion behaviors of SLM 316L and the wrought samples [25]. The PDP curves revealed that the SLM 316L samples exhibited a higher sensitivity to pitting corrosion in the aggressive solutions (1 M NaCl of pH 1 and 3 M NaCl of pH 3), whereas the wrought samples were more sensitive to the pitting than the SLM 316L samples in less aggressive solutions. Moreover, electrochemical impedance spectroscopy (EIS) and Mott–Schottky measurements showed that a passive film on the SLM 316L samples was more stable than the wrought samples at high positive film formation potentials.

Corrosion Behaviors of SLM 316 SS in Miscellaneous Solutions

Trelewicz et al. compared the corrosion behavior of laser powder bed fusion (LPBF) 316L samples to wrought samples in a 0.1 M HCl solution, and the considerably diminished passive range and higher passive current density of the LPBF 316L samples as opposed to the wrought samples were measured using PDP tests [1]. The non-equilibrium microstructures that formed during LPBF manufacturing were regarded as the main reason for the reduced corrosion resistance. The corrosion behavior of 316L austenitic steel processed by casting and SLM was investigated along with the effect of hot isostatic pressing (HIP) in a $H_2SO_4$ solution by Geenen et al. [26]. Because the corrosion resistance of the SLM 316L samples were poorer than the casting 316L samples, the HIP attempted to improve the corrosion resistance by reducing the amounts of inclusions, which involves the oxides or carbides, the cracks, and the pores. However, the HIP after the SLM manufacturing led to worse corrosion resistance due to no significant reduction in porosity, spheroidized lamellar Si-rich oxides, and increased grain sizes. Lou et al. studied the effects of heat treatments and externally applied cold work on the corrosion fatigue crack growth rates of LPBF 316L samples under various loading frequencies in a boiling water reactor (BWR) normal water chemistry (NWC) condition, which contained 2 ppm dissolved $O_2$ at 288 °C [27]. Overall,

the fatigue crack growth rates of the stress-relieved samples, which were exposed to 650 °C for 2 h in argon, were higher than the solution-annealed samples, which were hot pressed at 1150 °C for 4 h and 1000 bar in argon. The externally applied cold work also increased the fatigue crack growth rates in all the samples. These trends became conspicuous as the loading frequency decreased, which is shown in Figure 3. In addition, the authors also looked into the effects of cold work and porosity on stress corrosion cracking (SCC) in an NWC and hydrogen water chemistry (HWC) condition that contained 63 ppb dissolved $H_2$ at 288 °C [28]. The SCC growth rates of the LPBF 316L samples were compared with the wrought samples, which showed rates that were lower than the rates of the LPBF 316L samples, while the samples in the HWC environment had lower SCC growth rates than the samples in the NWC environment under all conditions. It was also found that 20% cold work by warm-forging at 200 °C decreased the SCC growth rates of the 316L samples, and the porosity increased the SCC growth rates. In addition, the effect of the Si-rich oxide inclusions along the grain boundaries on the SCC in the NWC and the HWC conditions was investigated [29]. The Si-rich oxides preferentially dissolved and appeared to accelerate the oxidation, which caused extensive crack branching. Therefore, the control of oxygen and the high-oxygen-affinity elements, such as Si during LPBF manufacturing, would be necessary to reduce the SCC. In a study by Prieto et al. [30], the 316L SS made by DMLS were first corroded in a 6 wt.% $FeCl_3$ solution through scan tracks; this is one of the inherent microstructural defects in the DMLS manufacturing process. After the DMLS-prepared specimens were heat treated, the corrosion did not follow an obvious pattern that was associated with the scan tracks. This would suggest that the heat treatment has a reduced susceptibility of the DMLS materials to the preferential corrosion on the scan tracks compared to the as-received materials. Even though the heat treatment changed the corrosion damage pattern and the shape of the formed pits, it did not significantly affect the corrosion behavior of DMLS 316L.

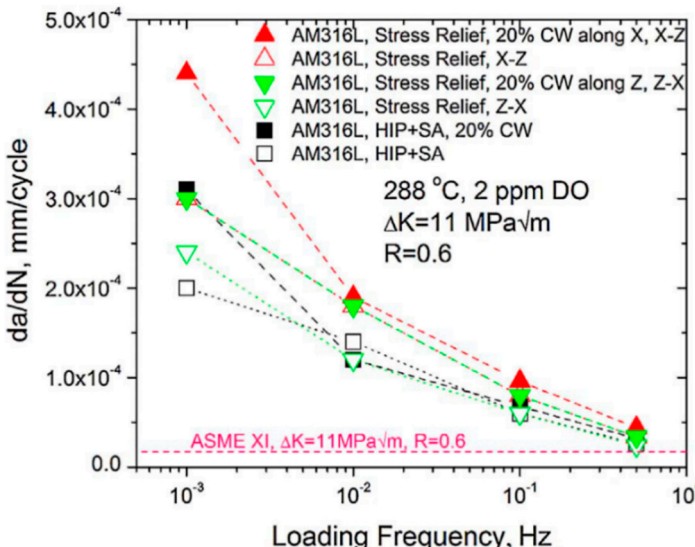

**Figure 3.** Comparison of the crack growth rate (da/dN) frequency between the cold-worked and the non-cold-worked AM 316L SS with different heat treatments [27].

Stašić and Božić examined the corrosion behavior of SLM 316L with an additional 1 wt.% NiB by measuring the weight loss in a 5% HCl solution [31]. The weight losses of sintering the 316L and the 316L-NiB samples were 1.2 and 1.3 g/m²·h, which indicated that the addition of NiB negligibly lowered the corrosion resistance of the base 316L material. SLM 316L as-manufactured samples, which were non-oxidized, and thermal oxidized samples, which were under an atmospheric condition of 700 °C for 150, 200, and 250 h, were prepared by Harun et al. [32]. They conducted PDP tests to compare the corrosion behaviors of the samples depending on the existence of the oxide layers

in Ringer's solution. The thickness of the oxide layer increased as the thermal oxidation time increased, whereas only the sample oxidized for 150 h showed a lower corrosion rate than the non-oxidized sample. The formation of the $Fe_2O_3$ and $Cr_2O_3$ layers was a primary factor for an improvement in the corrosion resistance, so the extended oxidation time triggered the $Cr_2O_3$ layer to break down and reduced the corrosion resistance. In order to compare the corrosion behavior and the biocompatibility of the SLM 316L samples in simulated body fluid (SBF) under different laser powers in between 120 and 220 W, Kong et al. conducted PDP tests before and after immersion, and they monitored the long-term corrosion potential for 288 h [33]. The corrosion potential of SLM 316L increased as the laser power increased, which was in line with the highest corrosion resistance of the SLM 316L samples that were fabricated at 220 W. The SLM 316L samples at 195 and 220 W also exhibited considerably higher biocompatibility than a quenched 316L sample due to the lower release of Ni ions that are toxic to our cells. In addition, the authors considered SLM 316L SS as bipolar plates, which are used to transport reactants/products and conduct electrons in proton exchange membrane fuel cells, and they investigated the effect of heat treatment on the corrosion behavior of the 316L SS [34]. The SLM 316L samples exhibited inferior corrosion resistance to the wrought samples in a 0.5 M $H_2SO_4$ solution with 50 ppm Cl and 2 ppm F ions. However, they showed that the recrystallization heat treatment, which involved a temperature ramp to 1050 °C in 30 min and soaking at 1050 °C for 30 min in argon that was followed by furnace cooling, could improve the corrosion resistance due to a more uniform structure, a thicker passive film, and an improved proportion of the chromic oxide in the passive film. Yang et al. studied the impact of an Au coating on the corrosion resistance of the SLM 316L SS used as bipolar plates of proton exchange membrane electrolyzer cells [35]. The bipolar plates were printed from SLM SS, and a thin film Au electroplating was employed on the surface of the bipolar plates. The Au-electroplated SS bipolar plates exhibited good corrosion resistance in an $O_2$-saturated $H_2SO_4$ solution. Lyczkowska-Widlak et al. conducted electropolishing (EP) on the SLM and commercial 316L samples in an EP solution that consisted of $H_2SO_4$ (35 vol.%), $H_3PO_4$ (60.5 vol.%), and triethanolamine (4.5 vol.%). The roughness and the corrosion behavior of the 316L samples were evaluated using a contact surface profiler and PDP tests in Ringer's solution [36]. The EP was effective, because the EP SLM samples showed low surface roughness and improved corrosion resistance compared to the untreated samples. It was concluded that the corrosion rate of the SLM 316L samples could be made lower than that of commercial 316L samples by using EP. Chen et al. prepared the SLM 316L samples with coarse, raw, and fine powder, and they compared their corrosion behavior using the PDP and weight loss tests in a 6.0 wt.% $FeCl_3$ solution [37]. The PDP curves showed that the SLM 316L samples from the fine powder exhibited a higher corrosion potential by 18.94%, 28.09%, and 67.33% in the XY-plane, the XZ-plane, and the YZ-plane, respectively, when compared with the other SLM 316L samples. Nevertheless, the corrosion currents and the rates of the three cross-sections on the SLM fine powder sample were higher than the rates of the other two samples, which were consistent with the actual weight loss. Song et al. studied the irradiation-induced microstructure and the irradiation-assisted stress corrosion cracking (IASCC) behavior of the LPBF 316L samples for the first time [38]. The irradiation was performed at a temperature of 360 °C for the proton irradiation with raster-scanning at a vacuum chamber pressure below $1 \times 10^{-7}$ torr. The constant extension rate tensile testing in an NWC showed that the HIP LPBF 316L samples had lower IASCC susceptibility than the stress-relieved LPBF 316L and the conventionally forged 316L samples. The intergranular corrosion (IGC) resistance of the SLM 316L samples was investigated using various microscopy analyses and a double-loop electrochemical potentiokinetic reactivation test (DL-EPR) in a 0.5 M $H_2SO_4$ + 0.01 M KSCN solution by Laleh et al. [39]. Figure 4b shows that the degree of sensitization (DOS) values and the ratios of the maximum current density in the reactivation loop ($I_r$) to the current density in the activation loop ($I_a$) in Figure 4a of the SLM 316L samples were lower than their commercial counterpart. Focused ion beam scanning electron microscopy, which is illustrated in Figure 4c–h, showed that

the IGC along the grain boundaries in the SLM 316L samples was less severe than in the commercial samples. The SLM 316L/Ag samples were printed with a mixture of the 316L powder and a 5 wt.% submicron Ag powder to enhance the corrosion resistance of the 316L SS by Quan et al. [40]. The corrosion potential of the 316L/Ag samples was more positive and the corrosion current density of the 316L/Ag samples was 25% lower than the SLM 316L samples in 0.5 M $H_2SO_4$ + 2 ppm HF, due to the higher standard electrode potential of Ag, whereas the pores that were formed by the weak interfacial bonding between the Ag agglomerates and the 316L SS matrix led to the lower breakdown potential of the SLM 316L/Ag samples compared to the SLM 316L samples. Woźniak et al. optimized laser energy density that ranged from 100 to 1333 mJ/mm$^3$ for SLM 316L based on the electrochemical parameters, such as the corrosion potential [41]. Even though a laser energy density of 1000 mJ/mm$^3$ endowed the highest breakdown potential, there seemed to be no apparent repassivation phenomenon. Therefore, they concluded that 600 mJ/mm$^3$ would be an optimal laser energy density, considering its relatively higher corrosion potential, its breakdown potential, and its repassivation potential in Ringer's solution.

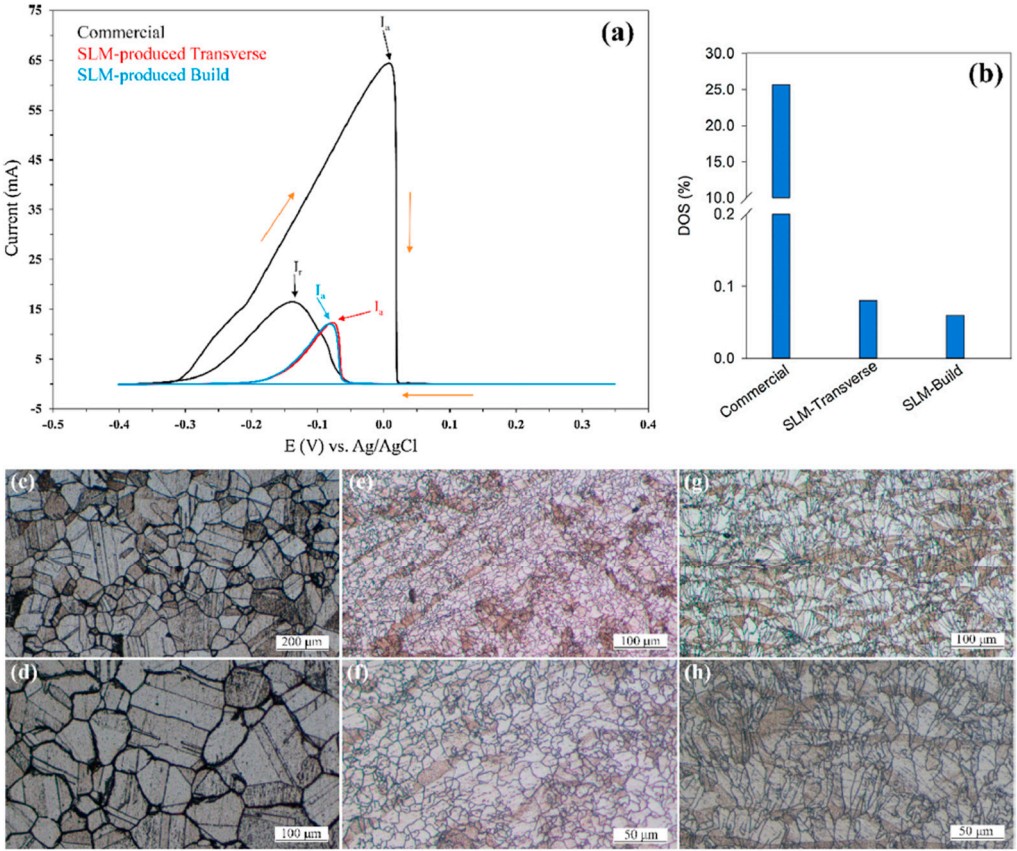

**Figure 4.** (**a**) DL-EPR curves for the commercial and the SLM-produced 316L samples; (**b**) the DOS values measured from the peak current densities of the forward and reverse scans in the DL-EPR curves; (**c**,**d**) the optical surface images after DL-EPR tests for the commercial 316L SS; (**e**,**f**) the SLM-produced 316L SS at transverse, and (**g**,**h**) at build [39].

### 3.1.2. Corrosion Behaviors of the SLM Miscellaneous Type SS
Corrosion Behaviors of the SLM Miscellaneous Type SS in NaCl Solutions

The corrosion properties of LPBF 17-4 PH were investigated by Schaller et al. [42]. LPBF 17-4 PH showed reduced corrosion resistance compared to the conventional wrought 17-4 PH materials based on the corrosion current density measurements. The micro-electrochemical cell experiments established that the presence of the pores on the sample surface directly affected the corrosion type, where active corrosion occurred above the

large pores, which had diameters $\geq 50$ µm, whereas the passive behavior appeared above the small pores, which had diameters $\leq 10$ µm. Stoudt et al. studied the effect of heat treatments on the corrosion properties of 17-4 PH SS that was manufactured by the SLM with nitrogen-atomized powder [43]. The PDP results confirmed that the pitting resistance was improved after the heat treatment, which consisted of homogenizing at 1150 °C and subsequent solutionizing at 1050 °C. The authors attributed the increased pitting potentials of the heat-treated samples to the finer martensite lath structure of the samples with more homogeneously distributed NbC precipitates than the wrought counterpart of the SLM samples as well as a stable passive film resulting from the absorbed nitrogen after the heat treatment. The influence of the re-austenization heat treatment on the corrosion properties of 17-4PH SS was assessed by Alnajjar et al. [44]. The re-austenitized SLM SS had a remarkably superior general corrosion resistance to the wrought SS. This was associated with the lower MnS inclusion content in the re-austenitized SLM SS. In the wrought SS, the dissolution of the MnS inclusions led to the deposition of a sulfur-rich layer in the adjacent regions, which promoted the destabilization of the passive film and thus deteriorated the general corrosion resistance. The X-ray photoelectron spectroscopy (XPS) measurements confirmed the formation of sulfur species on the surface of the wrought SS. Wang et al. compared an aging treatment with the combination of a solution and an aging treatment, which was referred to as the solution + aging treatment, to mitigate the corrosion of SLM 15-5PH SS [45]. The pitting potential of the samples with the solution + aging treatment was lower than that of the samples with the aging treatment. Moreover, the corrosion and the passive current densities of the samples with the solution + aging treatment were also higher than those of the aging treatment samples, which indicated that the stability of the passive film was weakened after the solution + aging treatment. Therefore, it was concluded that the samples after the aging treatment had a higher pitting potential and a more stable passive film than the samples after the solution + aging treatment.

Nath et al. studied the effect of heat treatments on the corrosion properties of the 420 SS fabricated by the LPBF [46]. There were no significant differences in the corrosion properties such as the corrosion current, polarization resistance, and corrosion rate, except that the pitting corrosion was improved after the heat treatment. Pateras et al. studied the effect of heat treatments on the corrosion properties of the SLM-processed 2205 duplex SS [47]. Considering the pitting breakdown potential and the hysteresis, the pitting corrosion resistance of the SLM-processed material was improved by heat treatment, which is shown in Figure 5. The pitting initiation and the hysteresis of the cyclic polarization curves in the studied potential range were observed in the SLM as-built samples, but they were not observed after the heat treatment.

Shang et al. studied the corrosion behavior of the UNS S32707 hyper-duplex SS with solution annealing from 1050 to 1200 °C [48]. The electrochemical experiments showed that the samples with a solution treatment at 1100 °C for 1 h had the best pitting resistance. The XPS measurements confirmed that the surface layers of the passive film were mainly composed of metal oxides/hydroxides such as Fe and Cr oxides. $NH_4^+$ adsorbed on the surface of the passive film protected the passive film. The inner layers of the passive film were mainly composed of metal elements, $Cr_2N$, and the oxides of Fe, Cr, and Mo. Hu et al. investigated the corrosion of S136 with additional $TiB_2$, which had wt.% of 0, 0.5, 1.5, and 2.5 [49]. S136 with 0.5% $TiB_2$ consisted of finer grains and fewer pores, contributing to the best corrosion resistance property. Figure 6 displays the surface morphology with low and high magnification of the pure SLM S136 and $TiB_2$/S136 composites samples after the PDP experiments. $TiO_2$, which formed from the corrosion process, acted as a new passive film that delayed the corrosion rate of the S136 matrix, so it had a positive effect on the corrosion resistance property. However, excessive $TiB_2$ preferred to form galvanic cells at the S136 matrix-$TiB_2$ interfaces, which was undesirable for the corrosion resistance property. Zeng et al. produced breathable mold steel that was produced by SLM from AISI 420 SS powder and 5% $CrN_x$ powder as a foaming agent [50]. The corrosion resistance of breathable mold steel was superior to the commercial PM-35 breathable steel, despite the

fact that the Cr content in PM-35 was higher than that in the breathable mold steel. The authors argued that the superior corrosion resistance was attributable to the enrichment of Cr around pore areas. Figure 7 shows the pore morphology of the as-printed 5% $CrN_x$ sample and the corresponding elemental distributions for Fe, Cr, and N. It was revealed that Fe distributed uniformly, whereas Cr and N were concentrated around the pore area. Shahriari et al. studied the anisotropy of the corrosion behavior in LPBF-fabricated SS CX by taking samples, which included side samples and top samples, in different regions of SS CX [3]. The side sample possessed a passive film with a more protective nature than the sample formed on the top plane, which contributed to the side sample's better corrosion performance, such as the higher charge transfer resistance, the higher corrosion and pitting potentials, and the lower corrosion current density. The microstructural differences, which included the residual stress, the dislocation density, and the low angle grain boundary density between the top and the side samples, contributed to the observed anisotropic corrosion behavior. Yang et al. produced 304 SS plates by SLM, and they then laser-welded the plates in three different directions to investigate the effect of anisotropy in the plate parts on their welding characteristics [51]. The corrosion resistance of the laser-welded joints was comparable to the wrought 304 SS, and it was superior to the SLM-produced base plates judging from the corrosion rate and the pitting potential. By comparing three kinds of laser-welded joints that varied from the welding directions, the authors argued that the anisotropy caused by SLM exerted a negligible effect on the corrosion resistance of the laser-welded joints. The corrosion behavior of SLM 304L was investigated and compared to the conventional wrought 304L by Schaller et al. [52]. Using electrochemical and immersion tests in 0.01, 0.6, and 1 M NaCl solutions, SLM 304L exhibited superior pitting resistance in a polished state compared to wrought 304L. However, the grit-blasted surface finish of the as-received SLM samples aggravated their exceptional corrosion resistance that was observed in the polished condition. The surface roughness and the embedded particles due to grit blasting introduced crevice-formers on the SLM sample surfaces, which reduced their localized corrosion resistance.

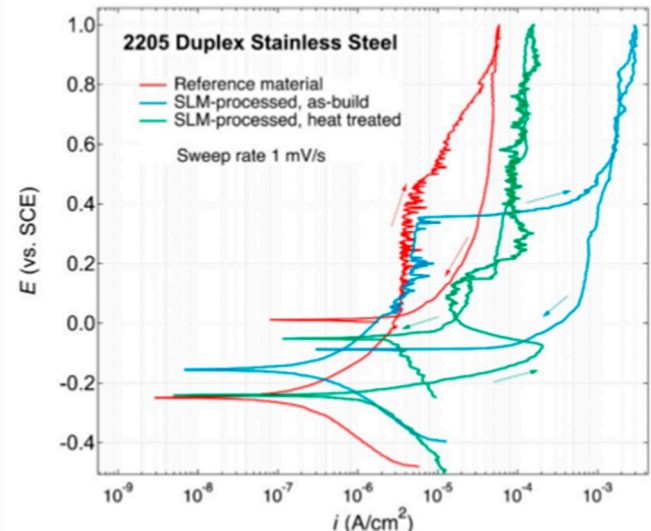

**Figure 5.** Cyclic potentiodynamic polarization (PDP) curves of the as-received cold-rolled/annealed reference alloy 2205, the as-SLM-processed duplex SS (DSS) 2205, and the SLM-processed DSS 2205 with heat treatment in a neutral 0.6 M NaCl solution at room temperature. The arrows indicate the polarization direction [47].

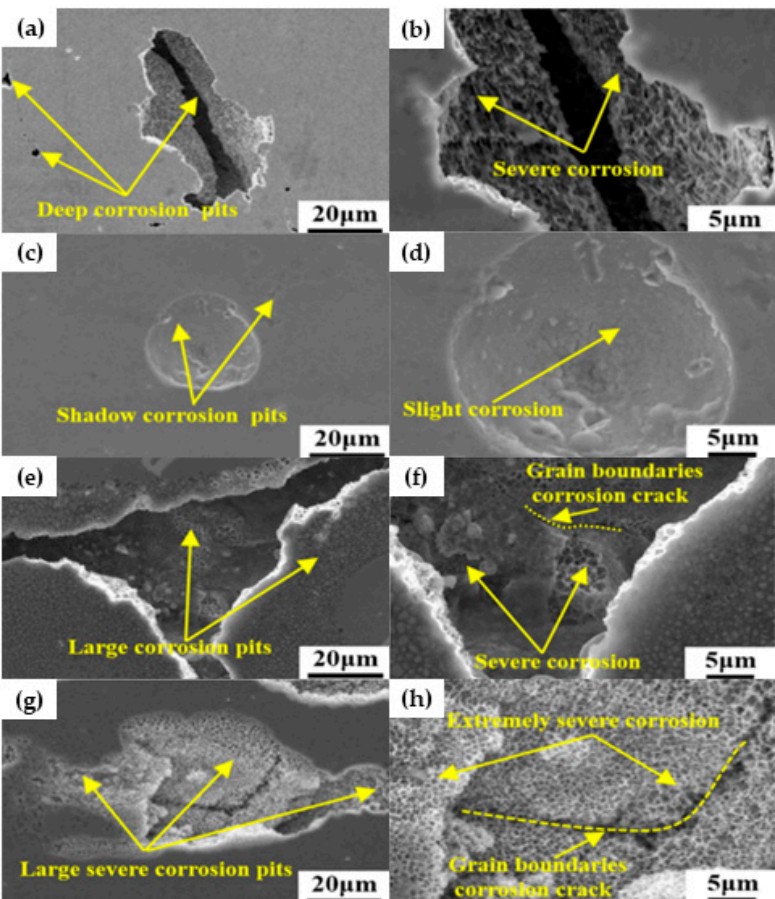

**Figure 6.** Surface morphology images of the SLM TiB$_2$/S136 composites after corrosion tests in the 3.5 wt.% NaCl solution: (**a**,**b**) pure S136; (**c**,**d**) 0.5 wt.% TiB$_2$; (**e**,**f**) 1.5 wt.% TiB$_2$; (**g**,**h**) 2.5 wt.% TiB$_2$. The low magnification is on the left, and the high magnification is on the right [49].

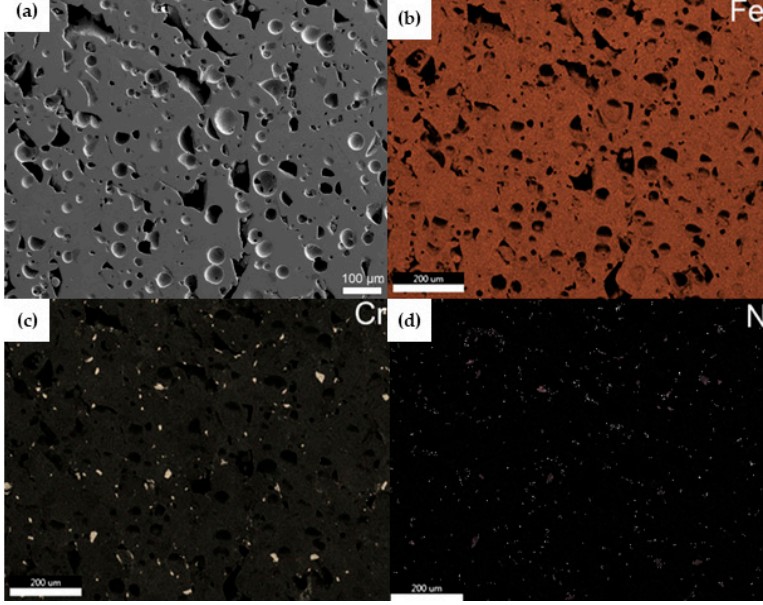

**Figure 7.** (**a**) Scanning electron microscopy (SEM) images of the 5% CrN$_x$ sample, and the corresponding elemental maps for (**b**) Fe, (**c**) Cr, and (**d**) N [50].

Corrosion Behaviors of the SLM Miscellaneous Type SS in Miscellaneous Solutions

Miller et al. compared the corrosion of LPBF AISI 316L with traditional manufacturing (TM) AISI 316L [53]. The LPBF and the TM samples were exposed to a 0.75 M sulfuric acid solution over 2184 h. While the mass losses of the LPBF and the TM samples were minimal, the surface roughness value of the LPBF samples increased drastically, which was contrary to the TM samples whose roughness values slightly decreased. The authors attributed this occurrence to the general corrosion on the LBBF samples. Schmidt et al. studied the corrosion behavior of LPBF AISI 4340 using electrochemical tests in artificial seawater [54]. Even though no discernible difference, which depended on the manufacturing conditions of samples, was found in the electrochemical tests, the sample surfaces showed different levels of localized corrosion attack in the form of pits. The samples with the "under sintered" printing parameters had the highest pit density and the largest size of pitting attack after the electrochemical tests. The authors argued that this could be due to the higher porosity as a result of the "under sintered" printing condition. Carluccio et al. investigated whether the SLM would be appropriate to produce pure Fe for biodegradable biomedical implants in Hank's balanced salt solution with an added 0.353 g/L of sodium bicarbonate [55]. The corrosion rate of the SLM samples was approximately 50% higher than the cast pure Fe. Therefore, the enhanced corrosion rate for easy biodegradation as well as the superior mechanical properties of the SLM samples would make SLM a promising manufacturing technology for biodegradable implants such as bone scaffolds. Similarly, Sander et al. studied whether SLM would be suitable for biodegradable implants [56]. They compared the corrosion of Fe–30Mn–1C–0.02S with a cast counterpart in the SBF. The SLM samples revealed a reduced corrosion activity compared to the cast samples. The authors argued that the SLM processing of this novel biodegradable FeMn-based alloy would be very promising for the efficient manufacturing of the complex implant structures. Peters et al. examined the FeMn-Ag alloys as potential bioresorbable implant materials made by the SLM from the mixed powder of FeMn and Ag [57]. The microstructure-dependent corrosion and the biomineralization processes in the SBF were studied in situ by means of the EIS, which clearly showed that the Ag-phases act as a local cathode within the FeMn matrix. However, the surface film formation was observed both for the Ag-phases and the FeMn-phases, which could potentially lower the self-corrosion as well as the galvanic coupling of the two phases. Figure 8 illustrates that both the FeMn alloy and the Ag substrate underwent pitting corrosion and the Ag surface was roughened substantially after the EIS experiment. While the pitting corrosion intensified in the FeMn-phase of the FeMn-Ag substrate, the surface morphology of the Ag-phase was hardly affected by the corrosion process, which would indicate that the Ag-phase in the FeMn-Ag substrate was cathodically protected by the actively dissolving FeMn-phase.

### 3.2. Corrosion Behaviors of the DLD SS

### 3.2.1. Corrosion Behaviors of the DLD 316L SS

In a study by Ziętala et al. [58], 316L LENS-manufactured steel had a similar corrosion potential (50.8 mV) to that of the classically manufactured surgery implant steel (40.0 mV). The breakdown potential and the repassivation potential were smaller, and the difference between the two values was greater, for the LENS-manufactured samples, which would imply that the pitting corrosion occurs much easier on the LENS-manufactured samples. Stull et al. reported that the DLD 316L SS clearly indicated a slightly lower corrosion resistance than the wrought 316L SS [59]. The corrosion resistance of the DLD 316L SS improved to some extent after the heat treatment, but not to the level of the wrought 316L SS. The corrosion characteristics of type 316L that was produced by DLD were investigated by Jung et al. [60]. They conducted a critical pitting test (CPT) and a crevice temperature test (CTT), as well as PDP on cast 316L and DLD 316L. Cast 316L had better pitting resistance than DLD 316L, and the passive film of DLD 316L was destroyed earlier than cast 316L. In addition, they measured the corrosion potential and the pitting potential after heat treatment on 316L at 1065, 1130, and 1200 °C. The difference between the corrosion

potential and the pitting potential increased as the heat treatment temperature increased, which showed the formation of a more stable passive film with an increase in the heat treatment temperature.

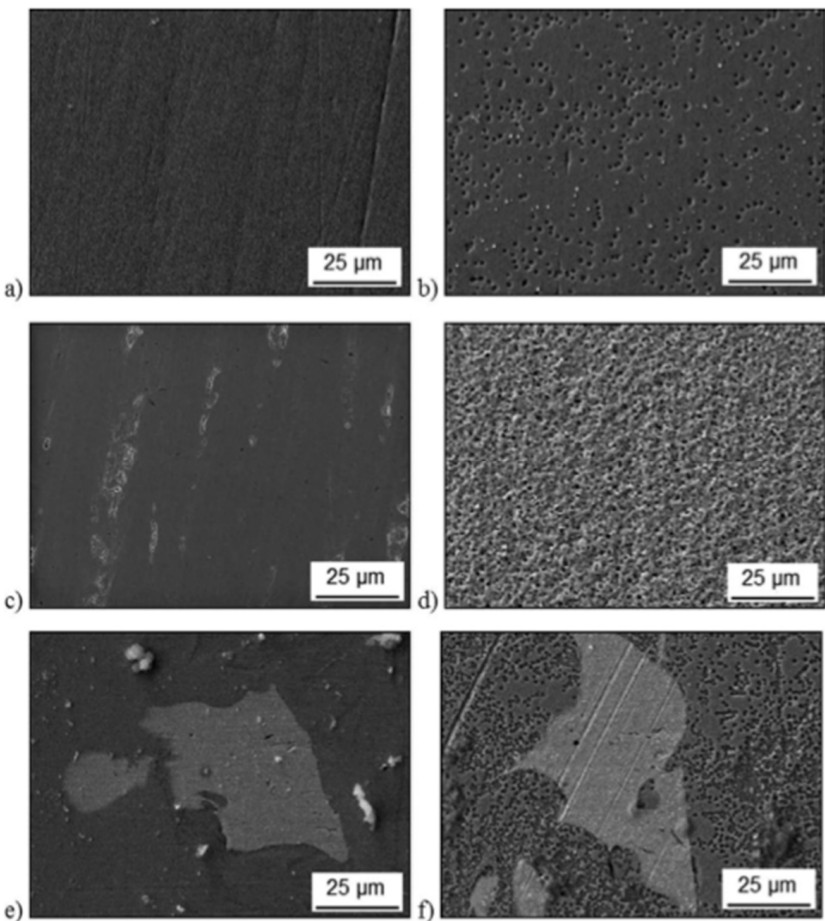

**Figure 8.** SEM images of (**a**,**b**) the FeMn sample; (**c**,**d**) the pure Ag sample; and (**e**,**f**) the FeMn-Ag sample before (left) and after (right) EIS [57].

### 3.2.2. Corrosion Behaviors of the DLD Miscellaneous Type SS

Other types of SS than 316L have been studied for corrosion. Lefky et al. prepared functionally gradient materials (FGMs) using DLD, and they correlated their Cr content with the etch rates in a 0.48 M HNO₃ and 0.1 M KCl solution [61]. The FGMs with varying numbers of tracks and layers were fabricated onto a 304 SS build plate. The first one to three layers of low chromium steel (LCS), which were followed by one to ten layers of 431 SS, are shown in Figure 9a,b. The authors evaluated the etch rate of the FGM samples, where the LCS region was electrochemically etched by the PDP. Incomplete mixing caused localized high Cr streaks within the first deposited tracks that resulted in slower etch rates. The mixing and the dilution of the LCS and the 304 SS were explained by the increasing presence of martensite with the increasing track number, which caused a faster etch rate. Additionally, the Cr concentration was related to the etch rate, which showed that even similar Cr tracks would result in different etch rates due to the microstructure changes as shown in Figure 9c,d. Melia et al. studied the corrosion behavior of the 304 SS in a 0.6 M NaCl solution [62]. They compared the DLD 304L to wrought samples, and the cyclic PDP indicated that the corrosion resistance of the DLD 304L was inferior to the wrought samples. It was shown that the presence of the lack of fusion (LOF) pores reduced the corrosion resistance by acting as crevice sites, and the corrosion resistance could be considerably enhanced by minimizing the LOF defects and by achieving a higher cooling rate. The

corrosion behavior of Fe-Cr-Ni-Mn-Mo-B steel was studied by Fang et al. They examined the effect of destabilization, which involved a heat treatment at 1050 °C for 30 min, on DLD samples in a 3.5 wt.% NaCl solution [8]. They compared the corrosion resistance of the as-deposited, the aged (which involved a heat treatment at 480 °C for 1 h), and the destabilized samples using cyclic PDP. The polarization behavior of the as-deposited and the aged samples was similar, but the corrosion resistance of the destabilized sample was superior to the other samples. The Cr content in the grain boundary region of the destabilized and the as-deposited samples of the Fe-Cr-Ni-Mn-Mo-B steel were 12.07 and 7.44 wt.%, respectively. Meanwhile, the same authors compared the corrosion behavior of Fe-Cr-Ni-Mn-Mo-Nb-Si steel with FV520B steel in a 3.5 wt.% NaCl solution [63]. The corrosion potential of the FV520B steel and the Fe-Cr-Ni-Mn-Mo-Nb-Si steel were −0.2830 and 0.0013 V, respectively, which indicated that the corrosion resistance of the DLD Fe-Cr-Ni-Mn-Mo-Nb-Si steel was superior to that of the FV520B steel. They attributed the difference in the corrosion behavior to a difference in the Cr content, and they confirmed that the Cr content in the DLD Fe-Cr-Ni-Mn-Mo-Nb-Si steel was higher than the other sample without the Cr-depleted zone. 35CrMo SS was produced with additional Ni, which included wt.% of 0, 3.05, 6.10, and 9.15, using LMD, and its corrosion behavior in 3.5 wt.% NaCl solutions was studied [64]. As the Ni content increased, the corrosion resistance improved, and the as-deposited specimen with 9.15% Ni presented the highest corrosion potential and the lowest corrosion current density. The polarization resistance and the corrosion rate of the 9.15% Ni sample was ~600% and ~10% of the no-additional-Ni sample.

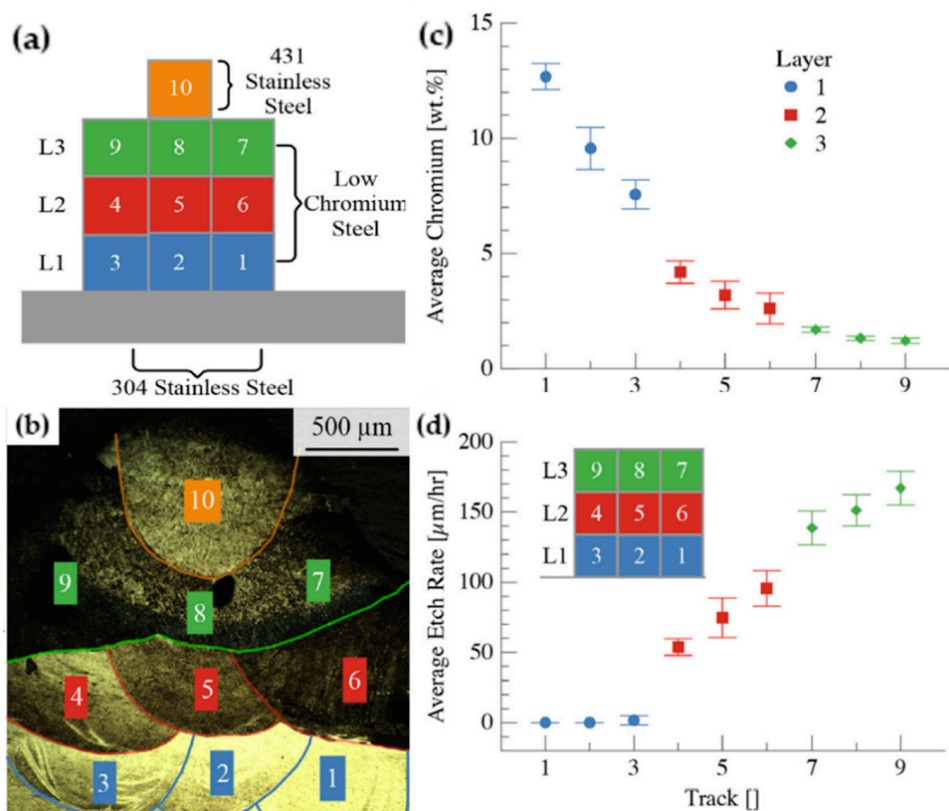

**Figure 9.** (**a**) Schematic diagram; (**b**) SEM image of the FGMs fabricated by DLD; (**c**) their Cr content; (**d**) the etch rate [61].

### 3.3. Corrosion Behaviors of the WAAM SS

The principle of WAAM is different from the other additive manufacturing methods that were previously described, and few papers regarding the corrosion study of WAAM-prepared SS have been published. Hao et al. used WAAM to prepare SUS304, and studied

a difference with the corrosion behavior depending on the types of applied current in the WAAM, which included a pulse current and a constant current [65]. It was indicated that the samples with a pulse current showed better corrosion resistance than the samples with a constant current. The authors attributed the improvement in the corrosion resistance on the samples that used a pulse current to the increased Cr and Ni content. Ron et al. studied the corrosion behavior of low carbon steel (ER70S-6), and the corrosion properties were evaluated by various methods such as salt spray and immersion tests [66]. The ER70S-6 samples were compared with their counterpart wrought ST-37 steel samples in an immersion test for 90 days and in a salt spray test for 45 days. The corrosion rates of the ER70S-6 and the ST-37 samples estimated in a PDP test were 0.08 and 0.05 mmpy. Overall, the corrosion resistance of the two samples was essentially quite similar, which was consistent with the immersion and the salt spray tests. Since a corrosion study of popular types of SS made by other additive manufacturing methods, which include 316L, is absent, a direct comparison of WAAM and other methods is difficult. Therefore, WAAM is an area where further research is needed in the future.

### 3.4. Corrosion Behaviors of the SS Fabricated by Miscellaneous AM Processes

In addition to the aforementioned AM processes, corrosion research has been conducted on the 316L SS manufactured by plasma transferred arc welding (PTAW), laser rapid manufacturing (LRM), and gas metal arc additive manufacturing (GMA-AM). The corrosion properties of the PTAW-produced and the LRM-produced 316L were investigated by Ganesh et al. [67]. The PTAW and the LRM samples showed lower resistance to pitting corrosion than wrought type 316L. Repeated exposure to the sensitization temperature range, ranging from 773 to 1073 K, during multiple layer deposition, which was coupled with a slower rate of cooling, resulted in the sensitization giving rise to continuous corrosion attack along the grain boundaries. In this case, the solution annealing of the LRM samples decreased the degree of sensitization with a small improvement in the pitting resistance. Chen et al. studied the effect of heat treatment on the corrosion properties of GMA-AM 316L SS [68]. The amount of $\sigma$ phase was reduced by increasing the heat treatment temperature and the time, which resulted in an improvement of the corrosion resistance. It is shown from Figure 10 that a pit nucleated, and the $\sigma$ phase was enriched at the interphase interface. A combination of 1200 °C and 4 h generated a complete austenitic structure that showed the best corrosion resistance. Zhang et al. examined the corrosion behavior of the Cr-Ni-based SS produced by laser cladding [69]. The as-fused cladding specimens developed pitting corrosion, which propagated in a horizontal direction, in 10 wt.% NaCl solutions. The as-fused cladding showed better corrosion resistance than AISI 1045 and the 304L SS in terms of corrosion potential, corrosion current density, anodic and cathodic Tafel slopes, polarization resistance, and the corrosion rate. Tarasov et al. looked into the corrosion behavior of the AISI 304 SS manufactured by electron beam wire-feed AM in an aqueous solution that contained 25% $H_2SO_4$, 5% $CuSO_4 \cdot 5H_2O$, and 12.5% NaF [70]. The $\delta$-ferrite content was determined by a heat input value, which was a critical variable that would control corrosion behavior, during the AM process, as shown in Figure 11a. The higher the $\delta$-ferrite content, the lower the corrosion resistance of the AM samples, which was due to the inefficient chromium oxide passivation. Figure 11b shows that a chromium oxide passivation film formed preferentially on the $\delta$-ferrite particles because the $\delta$-ferrite lattice was enriched by chromium as opposed to austenite. In sum, Table 1 classifies the relevant papers that are covered in this manuscript depending on the AM process, the SS type, the controlled parameter, and the relevant properties to corrosion behavior.

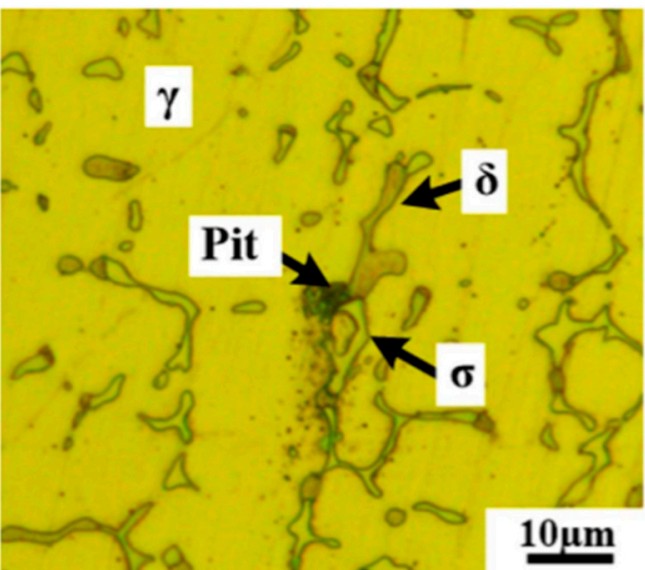

**Figure 10.** Pit nucleation at the interphase interface [68].

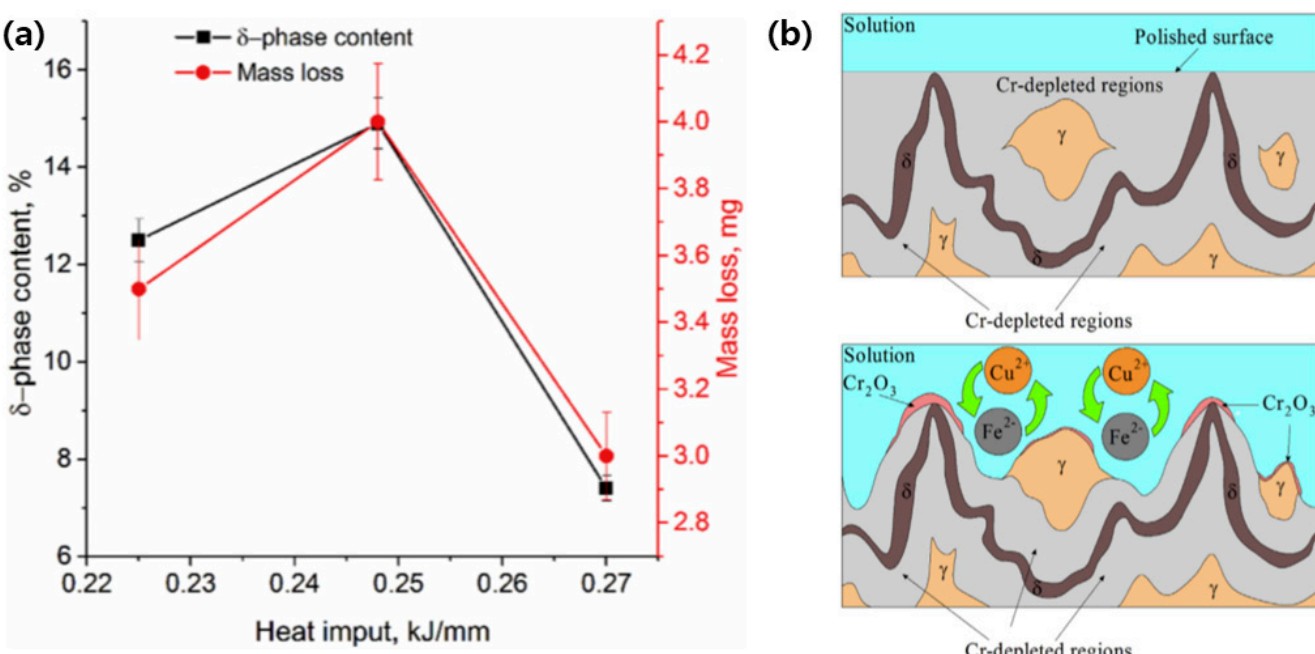

**Figure 11.** (**a**) δ-ferrite content and corrosion mass loss as functions of heat input; (**b**) schematics to show passivation film formation on δ-ferrite and austenite grains, as well as corroded surface morphology [70].

**Table 1.** Summary of relevant literature that specifies the controlled parameters and the relevant properties.

| AM Process | SS Type | Controlled Parameter | Relevant Properties | Corrosion Characterization | Ref. Num. |
|---|---|---|---|---|---|
| SLM | 316L | scan speed, laser powers | - | PDP, potentiostatic test | [15] |
| - | - | scan speed | - | PDP, OCP, EIS, linear polarization resistance | [16] |
| - | - | scan speed | closely-packed crystal planes (111), grain size | PDP | [17] |
| - | - | scan speed | porosity | PDP | [14] |
| - | - | heat treatments, cold work | - | corrosion fatigue crack growth tests | [27] |
| - | - | heat treatments | residual stress | PDP, EIS | [20] |
| - | - | heat treatments(thermal oxidized) | oxide composition | PDP | [32] |
| - | - | heat treatments | oxide composition | PDP | [34] |
| - | - | heat treatments | pit shape, corrosion damage pattern | PDP | [30] |
| - | - | cold work | porosity | PDP | [28] |
| - | - | HIP | - | SCC | [38] |
| - | - | HIP | porosity, si-rich oxide, grain size | PDP, OCP | [26] |
| - | - | laser powers | - | PDP, EIS | [33] |
| - | - | | - | PDP, weight loss | [37] |
| - | - | laser energy density | - | PDP | [41] |
| - | - | HPT plastic defarmation | cellular structure and voids | PDP, OCP, EIS | [21] |
| - | - | add TiC | - | PDP | [23] |
| - | - | Au coating | - | PDP, EIS | [35] |
| - | - | electropolising | surface roughness | PDP | [36] |
| - | - | NiB | - | weight loss | [31] |
| - | - | add Ag powder | pores | PDP | [40] |
| - | - | - | Non-equilibrium microstructure | PDP | [1] |
| - | - | - | Si-rich oxide | SCC | [29] |
| - | - | - | MnS content and presence of Cr deletion zones | PDP | [19] |
| - | - | - | lamellar microstructure | PDP | [22] |
| - | - | - | pores | PDP | [24] |
| SLM | 17-4 PH | heat treatment | microstructure, precipitates, passive film | PDP | [43] |
| - | - | heat treatment | MnS inclusion | PDP, OCP | [44] |
| - | - | - | pore | PDP, OCP | [42] |
| - | 420 | heat treatment | - | PDP | [46] |
| - | 2205 duplex | heat treatment | - | PDP | [47] |
| - | 15-5PH SS | solution treatment, aging treatment | passive film | PDP, OCP, EIS, potentiostatic pulse test | [45] |
| - | UNS S32707 hyper-duplex | solution treatment | passive film | PDP | [48] |
| - | S136 | add TiB$_2$ | passive film, grain size, porosity | PDP, OCP, weight loss | [49] |
| - | AISI 420 SS | add CrNx | distribution of Cr | PDP | [50] |
| - | SS CX | sampling direction | microstructural, passive film | PDP, EIS | [3] |
| - | 304 | welded directions | - | PDP, OCP | [51] |
| - | 304L | surface finish | surface roughness, embedded particles | immersion test, PDP | [52] |
| - | AISI 4340 | printing parameters | porosity | PDP, EIS | [54] |
| DLD | 316L | heat treatment | - | PDP, single loop-EPR | [59] |
| - | - | heat treatment | passive film | CPT, CCT, PDP | [60] |
| - | 304L | heat input | lack of fusion (LOF) pores | PDP, DL-EPR | [62] |
| - | Fe-Cr-Ni-Mn-Mo-B steel | heat treatment | Cr content | PDP | [8] |
| - | Fe-Cr-Ni-Mn-Mo-Nb-Si steel | - | Cr content | PDP | [63] |
| - | functionally gradient materials (FGM) | - | Cr content | OCP, PDP | [61] |
| - | 35CrMo | add Ni | - | PDP | [64] |
| WAAM | SUS304 | applying current type | content of Cr and Ni | PDP | [65] |
| LRM | 316L | solution annealing | sensitization | PDP, DL-EPR | [67] |
| GMA-AM | 316L | heat treatment | σ phase | OCP, PDP | [68] |
| electron beam wire-feed | AISI 304 | heat input | δ-ferrite | immersion tests, weight loss, flexural test | [70] |

## 4. Summary

We analyzed the corrosion behaviors of SS that depend on the AM process, the SS type, and the corrosion environment. Furthermore, we extracted the dominant experimental factors and the most relevant properties that affect the corrosion of AM-fabricated SS. In SLM 316L, the effects of the following parameters were investigated: the scan speed, the laser power, the laser energy density, and the post-treatment technologies, such as heat treatment and cold work. Changes in the scan speed, the laser power, and the laser energy density had an impact on the grain size and the voids of the samples, which were critical factors on the corrosion. In addition, the post-treatment affected the passive film stability, the repassivation potential, and the oxide composition of SLM 316L SS. Meanwhile, the effects of the heat treatment, the solution treatment, and the aging treatment were examined for the corrosion study of 17-4 PH, 420, 2205 duplex, 15-5PH SS, and UNS S32707 hyper-duplex manufactured by SLM. The pore, the microstructure, the precipitates, the MnS enclosure, and the passive film were correlated to the corrosion behavior. For the corrosion study of S136, AISI 420 SS, SS CX, 304, 304L, and AISI 4340 manufactured by SLM, some factors, which include the addition of $TiB_2$ and $CrN_x$, the sampling and the welded direction, the surface finish, and the printing parameters were considered, and it was confirmed that the passive film, the grain size, the porosity, the Cr distribution, the surface roughness, and the embedded parts affected the corrosion behavior. In DLD 316L, the most relevant papers focused on the effect of heat treatment on the passive film and the Cr content. There has been no specific trend in the corrosion studies of SS that are fabricated by the AM processes other than SLM and DLD. The controlled parameters in the corrosion study of SS are mainly the heat treatment, the printing parameters (the scan speed, the laser power, etc.), and the additives. The heat treatment is the most dominant factor that affects the corrosion behavior, regardless of the AM process for the production of SS. Interestingly, a comparison of corrosion behavior between SS that are produced by different AM processes with the same specifications has not been made. It is anticipated that more corrosion research in emerging AM processes, such as WAAM and droplet-based 3D printing, will occur in the future.

**Author Contributions:** Conceptualization, K.K. and T.-K.L.; methodology, G.K. and W.K.; writing—original draft preparation, G.K. and W.K.; writing—review and editing, K.K.; supervision, T.-K.L. All authors have read and agreed to the published version of the manuscript.

**Funding:** This work was supported by a National Research Foundation of Korea (NRF) grant funded by the Korean government (MSIT) (No. 2020R1F1A1053911).

**Institutional Review Board Statement:** Not applicable.

**Informed Consent Statement:** Not applicable.

**Conflicts of Interest:** The authors declare no conflict of interest.

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
