# Peer review of "The Corrosion of Stainless Steel Made by Additive Manufacturing: A Review"

_metals, doi:10.3390/met11030516_

Round 1
Reviewer 1 Report
The form of the article is typical for literature reviews. It is difficult to judge these types of articles because for the reviewer it is a secondary process. Someone has already reviewed the content of this article in the form of quoted publications.It does not bring anything new when it comes to research into materials produced by alternative methods. However, it is an interesting compilation of knowledge on the production and properties of materials produced by additive manufacturing. It can be useful for people dealing with similar topics as a kind of compendium of knowledge.
Author Response
Thanks for Reviewer 1's valuable comments.
Seemingly, Reviewer 1 doubts that the reviewing process of review papers summarizing literature in a certain field is appropriate or not. We think this is beyond the scope of revision of our manuscript. However, as Reviewer 1 points out, our review paper should be useful for people dealing with the corrosion properties of stainless steel manufactured by AM. We believe this is why review papers play an important role in science as well as original research paper.
Reviewer 2 Report
The authors present an interesting overview on a narrow, but important and further emerging field of manufacturing. Given the rising utilization of additive manufacturing of metal parts, durability and corrosion resistance will play an increasingly important role in the future, and a review on specifics for AM parts is providing a worthwhile basis for future works.
A key part of the review paper certainly is the summary table, which requires a few corrections:
- There is a numbering mismatch between the table and its in-text reference. (Table 1 / Table 2)
- Typo in the column headings, please correct to "relevant properties" (missing 's')
- Please correct all columns for consistent upper/lower cases
Moreover, the comparability of corrosion studies is ensured by using well-defined and standardized procedures. In order to gain a proper overview and understand the comparability of the described findings, the authors might want to add an additional column to their summary table, stating the standards utilized for corrosion measurements in the corresponding citations.
Reviewer 3 Report
Dear Authors,
I have read your paper "The corrosion of stainless steel made by additive manufacturing: A review" carefully.
This is an interesting article that tries to review the corrosion property of AM-fabricated stainless steel. The article is well organized and supported by current literature. The content is presented in a logical way and the final conclusions are well supported in the body of the article.
The paper requires a slight editorial correction.
Among other things, Line 54. Please, corrected the text. The P20 is not stainless steel.
Line 507. 304L
Please, add the chemical composition of the 625 alloy.
Line 513. Please, correct 625 SS. 625 alloy is the Ni-alloy.
Title
The article examined the stainless steel, but also the carbon, low-alloy tool steel. Perhaps it would be beneficial if the title "The corrosion of steel made by additive manufacturing: A review". I leave my suggestion to the authors for reflection.
After minor revision I can recommend the Editor to accept this revised manuscript to be published in Metals.
Reviewer 4 Report
This paper mainly presents a review work of the corrosion of stainless steel fabricated via different metal additive manufacturing technologies. This review is timely and meaningful. However, before being formally published in Metals, there are still some issues should be well addressed.
Some more detailed comments and/or suggestion are as follows:
- There are some grammar mistakes and/or typos throughout this paper. The English language of the revised paper should be double checked and polished by a native speaker. Moreover, the full name of SS should be presented at its first appearance in the Abstract.
- In the introduction section, the statement of “The AM adopts a bottom-to-top approach of rapidly sintering and melting powder layer-by-layer”is not rigorous, because not all the additive manufacturing technique use powder as the raw materials, such as wire arc additive manufacturing. This statement should be revised properly. Moreover, one branch of metal additive manufacturing, which directly deposits metal droplet, i.e., metal droplet printing, should be added and discussed in the introduction section. Some most recent work on this topic should be concluded, such as: Embedded printing trace planning for aluminum droplets depositing on dissolvable supports with varying section. Robotics and Computer-Integrated Manufacturing 63 (2020): 101898.; Toward 3D Printing of Pure Metals by Laser‐Induced Forward Transfer. Adv. Mater., 27: 4087-4092.; Direct fabrication of metal tubes with high-quality inner surfaces via droplet deposition over soluble cores. Journal of Materials Processing Technology 264 (2019): 145-154.; Effect of the surface morphology of solidified droplet on remelting between neighboring aluminum droplets. International Journal of Machine Tools and Manufacture 130-131 (2018): 1-11.
- The necessity and significance of conducting this review work should be further strengthened in the abstract and introduction sections.
- The introduction of principles of various additive manufacturing processes should be shortened and simplified, because these contents are not new and not the main scopes of this review.
- The schematic illustrations (4 kinds of additive manufacturing processes) shown in Fig. 1 and Fig. 2 are not typical and elaborated. Please improve the quality of these pictures.
- It is better to add some more contents and discussion on the comparative evaluation, especially for different technologies, in the discussion sections. Moreover, table 2 should be concluded in the main text rather than in the conclusions.
- Some more contents about future development trend should be added in the discussion and summary. Moreover, references should not be presented in the summary.
Round 2
Reviewer 4 Report
I thank the authors for their careful revisions, and I recommend this revised manuscript to be published in Metals.